# The ELECTRA Trial: Approach to Contemporary Challenges in the Development and Implementation of Double-Blinded, Randomised, Controlled Clinical Trials in Low-Volume High-Complexity Surgical Oncology

**DOI:** 10.3390/cancers17030341

**Published:** 2025-01-21

**Authors:** Sean Ewings, Nadia Peppa, Daniel Griffiths, Maria Hawkins, Claire Birch, Adly Naga, Georgina Parsons, Aymen Al-Shamkhani, Joanne Lord, Adrian C. Bateman, Andrew Bateman, Charlotte Lane, Kelly Cozens, Gareth Griffiths, Simon J. Crabb, Charles West, Hideaki Yano, Malcolm A. West, Alexander H. Mirnezami

**Affiliations:** 1Cancer Research UK Southampton Clinical Trials Unit, University of Southampton, University Hospital Southampton NHS Foundation Trust, Southampton SO16 6YD, UK; sean.ewings@soton.ac.uk (S.E.);; 2Department of Medical Physics and Biomedical Engineering, Faculty of Engineering Sciences, University College London, London WC1E 6BT, UK; 3Department of Medical Physics, University Hospital Southampton NHS Foundation Trust, Southampton SO16 6YD, UK; 4Patient and Public Representative, c/o Cancer Research UK Southampton Clinical Trials Unit, Southampton SO16 6YD, UK; 5Department of Immunology, Cancer Sciences Division, University of Southampton, Southampton SO16 6YD, UK; 6Southampton Health Technology Assessments Group, University of Southampton, Southampton SO17 1BJ, UK; 7Department of Pathology, University Hospital Southampton NHS Foundation Trust, Southampton SO16 6YD, UK; 8Department of Clinical Oncology, University Hospital Southampton NHS Foundation Trust, Southampton SO16 6YD, UK; 9Department of Radiology, University Hospital Southampton, Southampton SO16 6YD, UK; 10Southampton Complex Cancer and Exenteration Team, University Hospital Southampton NHS Foundation Trust, Southampton SO16 6YD, UK; 11University Department of Academic Surgery, Cancer Sciences, Faculty of Medicine, University of Southampton, Southampton SO16 6YD, UK; 12NIHR Southampton Biomedical Research Centre, Perioperative Medicine and Critical Care Theme, University Hospital Southampton, Tremona Road, Southampton SO16 6YD, UK

**Keywords:** surgical randomised blinded trial, intraoperative electron beam radiotherapy, locally advanced rectal cancer, locally recurrent rectal cancer

## Abstract

Designing and planning trials in surgical disciplines presents significant challenges, and many surgical treatments have not yet been evaluated with the highest standards of evidence. Nevertheless, achieving the high standards of randomised trials is a crucial goal, especially for financially constrained national healthcare systems, to ensure that funding decisions are backed by robust evidence. This challenge is further compounded in the context of low-volume, high-complexity, multi-specialty, and multi-modality interventions, which are increasingly employed in the management of some of the most difficult cancers. Here, we highlight some of the legitimate difficulties in designing such complex trials. Using the ELECTRA trial as an example, we suggest recommendations that may facilitate future surgical studies. ELECTRA is a randomised controlled, three-arm, double-blinded feasibility trial, testing the benefit of the addition of intraoperative electron beam radiotherapy at the time of surgery in the management of advanced and recurrent rectal cancers.

## 1. Introduction

### 1.1. Clinical Trial Challenges in Low-Volume, High-Complexity Surgery 

The application of high-quality research findings to drive clinical practice change is fundamental to delivering effective and safe clinical practice within an evidence-based management system. However, achieving this in surgery has always been challenging. Many aspects of common clinical practice have evolved not through high-quality evidence and randomised clinical trials (RCTs) but through lower-level studies that are susceptible to bias and confounding [1,2,3,4]. For instance, general surgical treatments were found to be only half as likely to be based upon RCT evidence than general medical treatments, and overall, the proportion of randomised clinical trials in surgery was observed to be low, representing 14% of research studies in the *British Journal of Surgery* in 1985, which further decreased to 5% in 1992 [5]. Indeed, in 1996, the then editor of the Lancet controversially stated that “…only when the quality of publications in the surgical literature has improved, will surgeons reasonably be able to rebut the charge that as much as half of the research they undertake is misconceived” [6].

This challenge is even more pronounced in the setting of low-volume but high-complexity surgical oncology. In this paper, we describe the approaches adopted in the design, development, and implementation of the Cancer Research UK ELECTRA trial, a randomised, controlled, double-blinded feasibility study in a low-volume, high-complexity surgical field, with a planned extension to a late-phase trial. ELECTRA is a three-arm clinical trial, testing the addition of intraoperativE eLECtron beam radioTherapy in locally advanced and locally recurrent Rectal cAncer coordinated by the Southampton Clinical Trials unit and sponsored by University Hospital Southampton NHS Foundation Trust.

### 1.2. Surgical Challenges in the Treatment of Locally Advanced and Locally Recurrent Rectal Cancer

To date, colorectal cancer remains a key public health issue, being the second leading cause of cancer-related mortality and of cancer-related healthcare expenditure [7,8]. The rectum is the most commonly affected site, and in the management of rectal cancer, perhaps the greatest therapeutic challenges have arisen in the management of locally advanced rectal cancer (LARC) or locally recurrent rectal cancer (LRRC). Uncontrolled pelvic disease from either LARC or LRRC has a very poor prognosis and causes severe morbidity and mortality, often described as one of the worst ways to die. Without treatment, median survival is typically 6–7 months and palliation is exceptionally difficult due to a growing tumour mass within the bony confines of the pelvis, leading to severe pain, often refractory to opiates, uncontrollable tenesmus, fistulas, malodorous discharge, and social isolation [9,10,11]. The best management option for LARC and LRRC is multimodality treatment often involving radical exenterative pelvic surgery, which offers the best chance for cure or long-term disease control. The most critical indicator of long-term oncological outcomes in such patients is the completeness of tumour excision by surgery. Accordingly, the most optimal outcomes observed occur in patients with a microscopically clear resection margin (R0), while those with clear but narrow margins or microscopically involved margins (R1) have significantly poorer outcomes and survival [12,13,14,15,16]. However, in the narrow confines of the bony pelvis, certain anatomical zones carry a significantly greater risk of incurring positive resection margins and, consequently, worse outcomes. Disease involving the lateral (pelvic sidewall) and/or posterior anatomical zones carries a much poorer prognosis due to the challenges posed by critical anatomical structures, such as the internal and external iliac vessels and their branches and tributaries, the sciatic nerve and its nerve roots, and pelvic bones. This makes achieving an R0 resection more challenging and increases the risk of catastrophic haemorrhage or major neurological morbidity [17,18,19].

Achieving an R0 resection for LARC and LRRC involving the sidewall or posterior pelvic compartments often necessitates aggressive multimodal treatment incorporating chemotherapy, radiotherapy, and radical and lengthy multi-visceral surgical resections. Despite significant incremental refinements in surgical techniques and optimisation of neo- and adjuvant therapies, achieving an R0 surgical resection remains challenging. Even in modern series and centres with significant experience in multi-visceral resections, positive resection margins may still occur in 30–50% of patients when the disease involves the pelvic sidewall and bony sacrum [20,21]. To further improve these outcomes, intraoperative electron beam radiotherapy (IOERT) has emerged as a potential additional and valuable modality, aiming to enhance local control.

### 1.3. The Potential Role of IOERT 

IOERT is defined as the direct application of high-energy electron beam irradiation to a tumour bed during an operative procedure [22,23,24]. This approach allows for the precise delivery of a single large fraction of radiation directly and specifically to high recurrence risk anatomical target areas, while simultaneously displacing and shielding dose-limiting radiosensitive structures, such as the small bowel or ureter, if they are not involved by the tumour. Consequently, the addition of IOERT to a tumour bed that has sustained an R1 surgical resection for LARC or LRRC may serve to mitigate for any residual cancer cells, potentially achieving an oncological outcome equivalent to an R0 resection [22,23,24]. Supporting this hypothesis, increasing evidence suggests that incorporating IOERT in the multi-modal treatment of LARC and LRRC can improve local control and survival. Several systematic literature reviews of low-quality studies have noted evidence of improvements in locoregional relapse in patients treated with IOERT [25,26,27]. As a result, international consensus recommendations have recently begun to consider the value of an IOERT boost in LARC and LRRC [28,29,30].

Despite the potential benefits, high-quality studies showing clear efficacy of IOERT have been lacking and their necessity has been emphasised by all consensus groups and national recommendations [14,15,24,25,26,27,28,29,30]. In addition, concerns have been raised about the cumulative toxicity of the added radiotherapy [25,26,27], especially in patients previously treated with pelvic radiotherapy. For example, patients with LRRC may have received up to 45–50 Gy external beam radiotherapy (EBRT) during treatment of their primary rectal cancer, followed by an additional 30 Gy of neoadjuvant reirradiation treatment upon diagnosis of an LRRC, and, finally, an IOERT boost of 10–15 Gy. This cumulative exposure raises valid concerns about radiotherapy toxicity. To date, there are no high-quality randomised, blinded studies evaluating IOERT in rectal cancer. The ELECTRA trial aims to address this gap. In this paper, we describe the challenges of designing and implementing such a trial and the collaborative process of trial development, including national and international consultations that were held to facilitate this. We outline the discussions to determine the best study design and defining the most optimal outcome measures to evaluate. We highlight the extensive patient participation and input into the study design. Additionally, given the study’s context in a complex, evolving field with no clear international standardisations, we describe the processes used to address internationally agreed-upon definitions, radiological standardisation, surgical learning curves, quality assurance, and pathological standardisation, along with the broader impact and benefits of these activities. Finally, we describe the novel design employed to facilitate the involvement of national and international units with varying levels of equipoise regarding IOERT. Considering that a recent study observed that one in five surgical trials are discontinued, while one in three completed studies remain unpublished, the development, design, and implementation of ELECTRA may provide valuable insights for future surgical trials [31].

## 2. Approach in ELECTRA and Trial Design

### 2.1. Multi-Level Stakeholder Participation and Input into Study Design

A key challenge identified at an early stage was the small niche field of pelvic exenteration surgery involving the pelvic sidewall and posterior compartment that may benefit from IOERT that ELECTRA pertains to. Since patient referrals depend on external units and any late-phase, practice-changing trial requires multi-institutional support for recruitment in such a small field, it was crucial to gain widespread clinical engagement and input from the diverse teams managing LARC and LRRC. Therefore, meetings and discussion groups were initially held at local, regional, national, and eventually international levels. These aimed to gauge the willingness to recruit patients and collaboratively identify and articulate the key questions, challenges, and mitigation strategies.

Following the positive outcomes of these discussions, multiple national and international meetings and workshops were conducted with invited groups of experts from the fields of radiation oncology, surgery, and physics, as well as methodologists. These were often held in conjunction with and as part of other established international meetings, such as the Association of Coloproctologists of Great Britain and Ireland (ACPGBI); the European Society of Coloproctology (ESCP); The American Society of Colon and Rectal Surgeons (ASCRS); The American Society of Clinical Oncology (ASCO); and the International Society of Intraoperative Radiotherapy (ISIORT).

These forums helped to articulate the following concerns about the prospect of such a trial:Identifying the best clinical question and target population;Identifying the most optimal trial design and addressing equipoise in the community;The challenges of obtaining funding and support for such a study;Addressing the lack of any patient and public involvement groups in this field;Challenges in the recruitment of patients in such a niche area;The role and mechanism of randomisation and blinding;Identifying the best outcome measures;Challenges in the field in general, with no widely accepted definitions and standards in the radiology, surgery, and pathological assessments of such tumours and specimens.

The outcomes of these meetings helped inform the key elements of trial design and implementation, and the important points above are explained in more detail in the sub-sections below. The full ELECTRA trial protocol however is available in detail elsewhere (www.southampton.ac.uk/ctu/trialportfolio/listoftrials/electra.page, accessed on 5 January 2025), and pertinent sections can be found in Appendix A.

### 2.2. Identifying the Best Clinical Question and Target Population

During the consultations, it became clear that IOERT was used in various units under different clinical settings for managing LARC and LRRC, leading to diverse opinions on what was suitable for clinical study. An early question was whether both LARC and LRRC could be studied in the same clinical trial. Some believed they were distinct conditions with different preoperative treatments, as LRRC patients often receive higher doses of external beam radiotherapy compared to LARC. However, others argued that since the surgical operation and IOERT treatment are similar, and the outcome measures would likely be comparable, both conditions could be included in the same trial. The early involvement of methodologists allowed for mitigation strategies, like stratified randomisation, which would balance any potential bias in the study arms. A consensus was reached that LARC and LRRC could be studied together.

Another question was how IOERT should be evaluated in the LARC and LRRC treatment pathways. Three potential questions that could be evaluated were identified:What is the role of IOERT in LARC and LRRC when added to standard of care neoadjuvant treatment and standard of care surgical intervention?What is the role of IOERT in LARC and LRRC when used as a replacement for external beam radiotherapy (EBRT) and intended to de-escalate EBRT doses?What is the role of IOERT in LARC and LRRC in modifying the surgical margins and potentially de-escalating surgical radicality?

Following discussions between clinicians and methodologists, it became clear through successive meetings and patient number modelling that units could potentially contribute to each question. Although the latter two questions were deemed the most clinically intriguing, they were also considered the most challenging for an initial study, especially for a newly formed interest group. Therefore, the first question was deemed the most suitable starting point. It was also acknowledged that if the first step proved successful, the other questions could be considered for future exploration.

A consensus was also reached that the most relevant population to study would be patients with a predicted close or positive resection margin based on imaging. IOERT, as a tool, is designed to mitigate against the possibility of a positive margin, which most commonly occurs in the sidewall and posterior compartments. Thus, predicted narrow or involved margins in these areas, as assessed by a specialist multidisciplinary team (sMDT), would be the key inclusion criterion.

### 2.3. Identifying the Most Optimal Trial Design and Addressing Equipoise

While phase 3 clinical trials are advantageous for achieving practice change, a key challenge identified early on was the potential recruitment concerns in such a low-volume, high-complexity field of surgery and radiotherapy. Early engagement with a Cancer Research UK Clinical Trial Unit with expertise in surgical trial design strongly supported the need for a feasibility phase to help determine whether some of the anticipated challenges could be overcome, followed by a larger late-phase study as the most efficient approach. Methodologically, it was also considered beneficial to design a feasibility stage that would allow the retention and transfer of data into the subsequent late-phase study to bolster patient numbers and statistical power. Consequently, ELECTRA was designed as a prospective, single-centre, double-blinded, randomised, controlled feasibility trial.

While the feasibility stage was planned as a single-institution study, it became apparent that multi-institutional and international support would be needed for a future late-phase study. Therefore, other institutions treating eligible patients were approached early in the study design to determine their potential participation in a future trial. These discussions revealed two broad types of institutions: those with long experiences of IOERT in the pelvis, and those with more recent experience. Institutions with long experience often felt they would not have equipoise to randomise to non-IOERT-containing arms, while units with more recent experience felt they could. Nevertheless, initial modelling suggested that recruitment to a late-phase study would ideally require contributions from both types of units. Consequently, design modifications to allow both such units to participate were discussed. It was subsequently felt that introducing a dose escalation arm would enable units with long experience of IOERT, who felt it unethical to randomise to non-IOERT arms, to contribute to study arms with different doses of IOERT, while units with equipoise for a non-IOERT arm would contribute to all the study arms. This approach was widely acceptable to all the groups in the discussions. As a result, the final trial schema is outlined in Figure 1. Patients would be randomised in an equal 1:1:1 ratio to receive either extended margin surgery alone, extended margin surgery and standard dose IOERT (10 Gy), or extended margin surgery and high-dose IOERT (15 Gy).

An additional crucial aspect of equipoise was considered to be patient equipoise. Prior to ELECTRA, nearly all the eligible patients and PPI groups had consistently shown a strong preference for the rationale behind IOERT and desired access to it as part of their multimodality care. However, methodologically, it was believed that allowing preferences to play a role could significantly impact the ability to recruit and retain patients in certain treatment arms, as patients might be unwilling to be assigned to arms they perceived as inferior, thus affecting the feasibility of any trial. Therefore, it was concluded that blinding participants would be essential for the trial’s success and for minimising potential bias. Consequently, another aim of the feasibility stage was to determine the acceptability of randomisation and blinding for patients, as well as retention.

### 2.4. The Challenges of Obtaining Funding and Support for Such a Study

Once the initial outline of the study was established, funding bodies, charitable organisations, and industry partners were approached to assess the potential for support and funding. Examples of funding bodies included within the UK were the National Institute of Health and Care Research (NIHR); Cancer Research UK (CRUK); the Medical Research Council (MRC); the PLANETS Cancer charity; and Bowel Research UK, while externally, engagement was also sought from IntraOp, the manufacturer of one of the mobile linear accelerator devices used for IOERT.

A key aim of these initial discussions was to gauge the interest in such a study, as a lack of traction at this stage would clearly indicate that the study might not be suitable for further evaluation. In addition, recognising that any future late-phase study would require multi-centre and likely international recruitment, options for international funding were explored early on.

Positive discussions with funding bodies for a UK study allowed the proposal to progress, and suggestions for the funding of an international study were also optimistic. Options for such an extension depended on success in the UK and included funding the entire international effort through one national funding body, separate funding for other units through equivalent national funders, or finding suitable funding organisations or industry partners willing to fund the entire international study’s requirements.

### 2.5. Addressing the Lack of Patient and Public Involvement Groups

Patient and public involvement (PPI) in clinical studies can significantly enhance their effectiveness, importance, recruitment rate, and overall impact. Early-stage input from PPI can particularly aid in designing clinical studies to ensure they are more patient-centred, with relevant research questions, well-crafted information sheets, and peer advocacy [32,33]. Due to the rarity and high specialisation of this field, there were no established PPI groups available at the time to assist in this process. Therefore, patients from various institutions were initially approached for their interest during the design stage. Face-to-face and online discussions were conducted to gather input and opinions, with participants being remunerated for their involvement. The PPI working group contributed to all aspects of the study, including reviewing study documentation, assessing complications and risks from IOERT, and evaluating the methods, design, and approaches for the acceptability of randomisation.

Once the design was established, a smaller group of PPI members was formed to be part of the trial management group and assist in conducting the study. From this beginning, broader groups of patients have been involved, and support networks and online forums have developed. These forums, both closed and open, encourage active discussions and real-time feedback, often hosted by regional charities. The study design has also been shaped by several public engagement activities for eligible patients. Importantly, these events were attended not only by patients with LRRC or LARC but also by their family members and other members of the regional patient cohort with other subtypes of colorectal cancer, representing a broader spectrum of patient and public stakeholders. Our intention was that the presence of PPI representatives would lay the groundwork for a PPI panel to work on aspects of any future later-phase trial, communicate with lay audiences, and identify and prioritise topics important to this patient group. The PPI team also had trial oversight through the Trial Steering Committee.

### 2.6. Challenges in Recruitment of Patients in Such a Niche Area

One concern repeatedly raised during the development stages was the anticipated low recruitment in this highly specialised field. However, the regionalisation of complex cancer care, which leverages the volume–outcome relationships in surgery, has mitigated this concern to some extent. Initial modelling suggested that projected recruitment might be sufficient. Nevertheless, a feasibility stage assessment of this outcome measure was deemed necessary. If successful, this would provide reassurance for success in any future late-phase study. Due to the scarcity of institutions offering both this type of surgery and IOERT and the lack of other IOERT-ready units in the UK, it became clear that any future trial would require international support. Initial discussions with both high- and low-volume IOERT-capable institutions during the development phase supported this need.

### 2.7. The Role and Mechanism of Randomisation and Blinding

Extensive pre-trial discussions raised concerns about potential bias in such surgical trials. For example, it was considered that if the surgical team knew in advance whether IOERT was to be delivered, it might influence the radicality of surgery. Extra margins might be taken in cases where IOERT was not applied, or conversely, surgical margins might be reduced if IOERT was known to be delivered, thereby biasing outcomes and confounding interpretation. As a result of these discussions, blinding the surgeon and oncologist was deemed a critical bias-reducing design element introduced into ELECTRA.

The process was designed so that during the procedure, the surgical team would conduct the exenterative procedure as intended and directed by the MDT-determined surgical roadmap. Subsequently, the IOERT team, composed of a clinical oncologist, a medical physicist, and radiographers, would be called to the theatre. At this point, the surgeon and clinical oncologist would assess the specimen, the tumour bed, and preoperative imaging to determine the utility of IOERT. If IOERT was deemed necessary, the applicator would be positioned, and the system set up. The theatre would then be vacated (with remote anaesthesia and monitoring operational as standard), and patients would be randomized at this stage via a web-based system (1:1:1 ratio) by the lead physicist, who would be the only individual to know the outcome of the randomization, i.e., whether IOERT would be administered and at what dose, or if no IOERT would be delivered. The surgeon, oncologist, and patient would remain blinded throughout the study, and randomisation would be stratified by LARC or LRRC, with the maintenance of blinding being an outcome measure of the study. Nonetheless, conditions for unblinding were planned in the event that any participant’s further treatment might benefit from additional radiotherapy, and if potential IOERT treatment and its dose might therefore impact this.

### 2.8. Identifying the Best Outcome Measures

#### 2.8.1. Primary Outcomes at Feasibility Stage

The aim of the feasibility stage was to determine if a future phase II/III study on IOERT in LARC and LRRC is possible and to test the methods to be applied in pragmatic, real-world settings. Consequently, the primary outcomes at feasibility in ELECTRA are as follows:Number of patients meeting eligibility criteria and number of patients referred to a specialist MDT over the trial period;Number of patients accepting randomisation;Number of patients for whom IOERT was successfully delivered as planned;Number of patients and clinicians for whom blinding was maintained during IOERT delivery;Percentage of patients whose questionnaires can be analysed.

As a result, it was determined that meeting the aforementioned criteria, coupled with adequate community support, would render a future late-phase study in this field feasible.

#### 2.8.2. Secondary Outcomes at Feasibility Stage

Secondary outcomes were designed to evaluate whether the proposed data collection methods for assessing effectiveness and cost effectiveness endpoints are feasible and appropriate for this population. Additionally, they aimed to obtain pilot oncological, quality of life, and health economic data on patients treated with or without IOERT as part of a prospective blinded, randomised study to inform future work.

Secondary outcomes in ELECTRA include the following:Morbidity (Clavien–Dindo) at surgery and at 30 days post-randomisation;Mortality at 30 days and 90 days;Oncological outcomes: IOERT field recurrence, overall local recurrence, and overall survival (at minimum 12 months post-randomisation);Treatment-related toxicity (at minimum 12 months post-randomisation);Time to local or systemic recurrence;R1 rate;Quality of life information on the groups;Resource use and costs.

#### 2.8.3. Identifying and Defining the Best Outcome for Future Late-Phase Trial

The key goal of IOERT is to avoid re-recurrence in margin-concerning cancers by mitigating for a possible microscopic R1 resection through the use of a radiotherapy boost directed at residual clones. Consequently, the most important primary outcome measure of IOERT efficacy for any future late-phase study is IOERT field local control. However, at the time of this study’s design, there was no optimal method for accurately charting the intraoperative IOERT field for future surveillance, and IOERT field evaluation was poorly reported [25]. As a result, a method of marking out the IOERT field by applying radiopaque surgical clips to the margin of the IOERT field with the applicator still in situ (Figure 2) was devised. This simple but novel technique allows for precise tracking of the IOERT field postoperatively in sequential scans to report any evidence of IOERT field local recurrence or out-of-field local recurrence.

### 2.9. Challenges in the Field with No Widely Accepted Definitions and Standards in the Radiology, Surgery, and Pathological Assessments of Such Tumours and Specimens

During the design stage, several key elements were identified that would require standardisation and quality assurance in any future late-phase, practice-changing study, which would need testing and development at the feasibility stage. These included the standardisation of preoperative imaging to evaluate the key surgical margins, the description and recording of the surgical technique and extent of radicality, consideration of entry criteria for surgeons, methods for evaluating surgical technique and key outcome measures, and the standardisation of pathological reporting of the specimen for margin status. Consequently, working groups were established to discuss and develop these elements. The output standards for these are provided in Appendix A.

In the field of pelvic exenteration surgery, there was no consensus on the extent of the radicality of the surgical intervention or an appropriate classification system. Pelvic exenteration and beyond TME surgery are broad umbrella terms used for surgical interventions for LARC and LRRC, encompassing significant heterogeneity. Addressing this was an important first step. The UK Pelvic Exenteration Network (UKPEN) represents an association and group of institutions and individuals with a strong interest in this field and has been developing tools in this space. The UKPEN lexicon for complex pelvic cancer surgery has recently been developed and was used in ELECTRA [34], representing the first clinical trial in this space to adopt this system. For the assessment of surgical technique and quality, surgical photographs of the pelvis after specimen extraction were considered a suitable starting point for assessing surgical outcomes and are used in ELECTRA. These, combined with operative photographs during the application of the IOERT, also form a useful assessment of the treatment field and assist in the reporting of IOERT field local recurrence.

Radiological standardisation applied in ELECTRA was previously developed in several UKPEN institutions and is presented in Appendix A.

Histopathological handling of specimens following pelvic exenteration represented another area of unmet need with no clear international standards identified. These specimens are some of the most complex sent to pathology departments, often containing major vascular and/or bony elements, making orientation and processing difficult. As the assessment of the R stage is critical in evaluating the need for IOERT, an important objective of the ELECTRA feasibility trial was to establish rigorous and standardised criteria for the preparation and evaluation of the resected pelvic exenteration specimen. These were developed as part of the workshops and stakeholder meetings and are set out in Appendix A.

## 3. Discussion

Historically, randomised clinical trials have not been the standard mechanism for evaluating surgical interventions due to the practical and methodological challenges they pose in the craft specialties. This issue is particularly pronounced in the setting of high-complexity, low-volume surgical interventions. Nevertheless, the RCT design is regarded as the “gold standard” for evaluating clinical interventions by many investigators and national healthcare bodies, especially in times of decreasing health budgets and financial constraints. Here, we describe the design and development of the ELECTRA trial to outline some of the obstacles one may encounter in contemporary trial design in low-volume, high-complexity surgical oncology and make recommendations on how such obstacles may be overcome.

ELECTRA is designed to determine the incremental benefit of adding IOERT to the treatment of LARC and LRRC where margin control is a concern. Although the rationale for using IOERT appears sound, and safety elements have been addressed in previous studies, there is a lack of high-quality evidence on its efficacy. Consequently, ELECTRA was devised as a prospective, randomised, double-blinded feasibility trial with subsequent late-phase progression to evaluate this.

In our view, a crucial aspect is the early, extensive, collaborative, and iterative stakeholder engagement. This approach facilitates a well-thought-out and carefully designed study. We believe it enhances patient-centred approaches; makes excellent use of the wider community; builds collaboration, consensus, and trust; and establishes the relationships necessary for trial success. Esmail and colleagues noted that this method could improve the dissemination and uptake of results and approaches, thereby enhancing the quality of studies [35]. Adhering to this principle also helps define the best clinical question to address and the optimal target population through consensus. Critically, it assists in designing the most effective trial and determining the level of equipoise within the community of investigators and patients.

A potential obstacle in surgical studies is the lack of equipoise. During our pre-trial workshops, we observed that the IOERT community includes institutions with a long history of IOERT delivery and newer institutions that have recently adopted it. The former may sometimes lack equipoise in this field and express concerns about randomising patients to a non-IOERT arm, as their standard of care, developed over many years, includes IOERT treatment in the multimodality management of LARC and LRRC. The latter may not have a clear “standard” of care and thus may have greater equipoise in randomising to non-IOERT arms. In designing ELECTRA and considering future late-phase studies, it was important to create a novel study that would allow participation from both types of institutions. Consequently, a three-arm approach was devised, enabling newer institutions with equipoise to recruit to non-IOERT arms and participate in all three arms of the study, while allowing institutions unable to ethically recruit to a non-IOERT arm to recruit to the dose escalation IOERT arms only.

A further important obstacle is obtaining funding. Funding should be considered early, and advice from funding bodies at this stage of early design can greatly enhance the success of applications later, while also providing valuable informal study reviews.

Perhaps two of the most critical concerns in surgical trials are patient recruitment and retention, particularly in low-volume, high-complexity surgical oncology. Recruitment was identified as the key factor in the discontinuation of more than 20% of all surgical trials, according to a review by Chapman and colleagues [31]. In the ELECTRA trial, we addressed these concerns in several ways. The centralisation of complex surgical services has allowed units to serve larger populations, greatly aiding the process. By developing a supra-regional complex cancer MDT and facilitating collaborative and accessible discussion of all advanced and complex cancers needing radical surgical interventions within this MDT, we enhanced the discovery of eligible patients further and routed them towards ELECTRA for eligibility assessment. To reduce patient withdrawal due to a lack of equipoise and enhance retention, we opted to blind the participants (in addition to the clinicians). Although patient choice is crucial in developing new treatments, our initial surveys of patients and the public indicated that most patients preferred receiving IOERT, which would have impacted ELECTRA’s feasibility. To report on this key patient preference, we also recorded it as an outcome measure.

Blinding and randomisation approaches are critical to reducing several forms of bias but can be notoriously difficult to achieve in surgical studies. In ELECTRA, our pre-trial workshops suggested that blinding and randomisation were possible and desirable, and so we were able to design a novel method of keeping the treating surgeon and oncologist both blinded to the intervention received. Thus, by ensuring the surgery was conducted without knowledge of IOERT delivery and by introducing randomisation conducted by the attending medical physicist, in the absence of either surgeon or oncologist, we were able to maintain clinician blinding to the treatment received.

A further challenge within the field of LARC, LRRC, and IOERT is the lack of standards and definitions. In a previous study of EORTC protocols of surgical studies, Tanis and colleagues noted that robust quality assurance initiatives were rare in most surgical studies evaluated. Important elements of surgical quality, including definitions, credentialing, technique, and margins, were inconsistently and poorly described [36]. This issue is particularly pronounced in pelvic exenteration surgery for LARC and LRRC, where there are no clear definitions of the surgery’s magnitude and no standardisation of radiology or pathological assessment. In ELECTRA, we addressed this both actively and passively. Passively, the workshops and meetings that were held led to the unplanned but highly welcome formation of additional discussion groups, facilitating the development of standards and definitions in the field. Actively, these meetings resulted in clearly articulated standards of radiology and pathological assessment that the teams felt were acceptable and could be adopted.

Finally, we believed that in a challenging area of clinical study, a robust feasibility stage would be an essential safeguard and investment providing a realistic assessment of the capacity and capability to conduct any future trial with the objectives of informing on recruitment, acceptability, timelines, and cost.

Despite the efforts described, ELECTRA is still subject to certain limitations. The ability to recruit patients remains untested and an area of ongoing concern, and the ability to adopt and apply the standards described for radiology, surgery, and pathology robustly and routinely also remains a concern. Nevertheless, we believe that the process of describing and documenting these as part of a trial protocol will be advantageous in progressing the field. Lastly, if feasibility is achieved, a crucial issue will be advancing the ELECTRA trial to the study phase. Sustaining momentum and obtaining the appropriate resources to achieve this will be critical to determining the incremental value of IOERT in the field of LARC and LRRC.

## 4. Conclusions

The ELECTRA study represents a complex, innovative trial design addressing an unmet need in a niche area of high-complexity work. As the first randomised study in the field of pelvic exenteration, it is hoped that the measures taken towards standardisation and quality assurance in radiology, surgery, and pathology for these complex cases will pave the way for future pragmatic late-phase studies in this challenging clinical field. Using ELECTRA as an example, we have highlighted some of the genuine challenges that may be encountered in designing trials for high-complexity, low-volume interventions. We also provide recommendations on elements that may help overcome these pitfalls, while enhancing participant recruitment and retention.

## Figures and Tables

**Figure 1 cancers-17-00341-f001:**
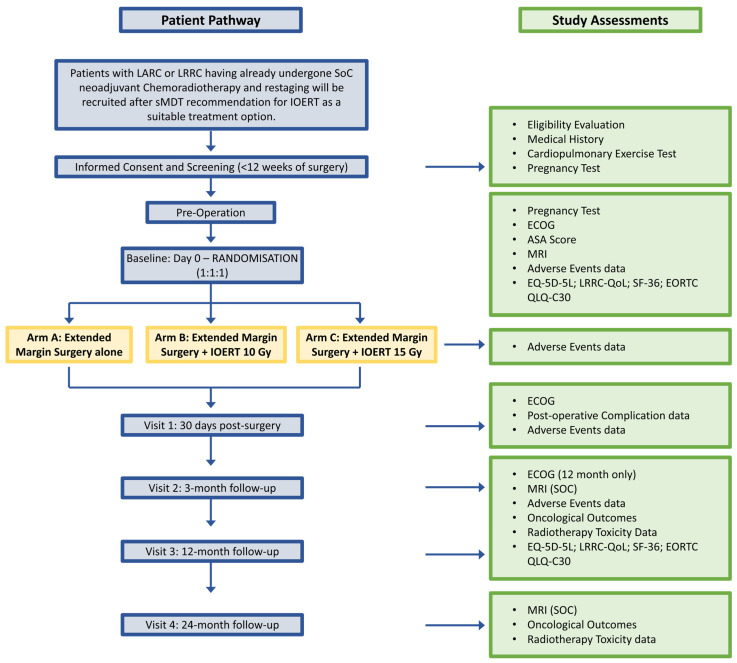
Electra trial schema. SoC, Standard of care; sMDT, Specialist Multi-disciplinary Team.

**Figure 2 cancers-17-00341-f002:**
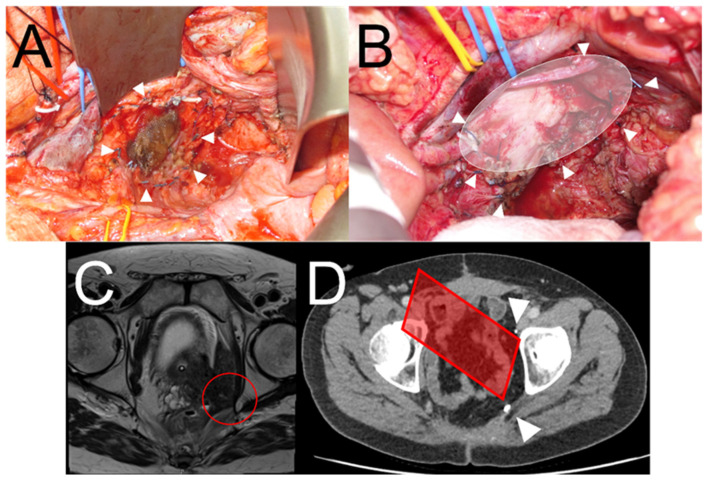
(**A**) View of the pelvic sidewall following pelvic exenteration and resection of pelvic sidewall including internal iliac vessels. The arrowheads show the application of metallic clips to the IOERT field for future tracking. (**B**) Similar view following pelvic exenteration with en bloc resection of the pelvic sidewall vessels exposing the sciatic nerve. The arrowheads and schematic oval shape show the site of the IOERT field. (**C**) Preoperative MRI of patient needing pelvic exenteration who was potentially eligible for the ELECTRA trial. The red circle shows the area at greatest risk of an R1 surgical resection. (**D**) Same patient in (**C**) postoperatively who received IOERT. The arrowheads show the ability to track the IOERT field with diagrammatic representation of the IOERT applicator as it would have been placed per-operatively.

## Data Availability

No new data were created or analysed in this study. Data sharing is not applicable to this article.

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
