# Peer review of "The ELECTRA Trial: Approach to Contemporary Challenges in the Development and Implementation of Double-Blinded, Randomised, Controlled Clinical Trials in Low-Volume High-Complexity Surgical Oncology"

_cancers, 2025, doi:10.3390/cancers17030341_

Round 1
Reviewer 1 Report
Comments and Suggestions for Authors
Thank you very much for letting me review this interesting paper.
The authors address an important subject, the difficulty in executing randomized trials in low volume complex oncologic surgery.
Es an example, the authors describe their experience in the conception of the Electra trial.
Here, the authors address several difficulties in the conception / preparation of this study and its study design and how to overcome these.
Here are my comments:
- For this specific topic, IOERT in LARC/LRRC, a randomized controlled study design was possible to practice. This might not be possible in other clinical settings where operator - blinding is not possible, for example when different medical advices have to be used and the operator has to know which one to use.
- in my opinion the introduction is too long and should be shortened if possibile to 1.5 pages.
- The authors describe their experience of interdisciplinary discussions on a regional, national and international level: In my opinion, some word should be spent about big data. In modern times, especially when it comes to research in rare cases, physicians / insurance companies and especially national healthcare systems need to have an interest in performing research with big data connecting multiple institutions in order to accumulate patients.
What are the intentions of the authors concerning these points? Did the authors opt for multi center approaches for the Electra trial?
Author Response
Comment 1: in my opinion the introduction is too long and should be shortened if possible to 1.5 pages.
Response: Thank you for your feedback. We have addressed it in the revised manuscript, abridging several sections, mainly in the first subheading. The challenge of shortening the text was that there are several key concepts that need to be introduced to understand the difficulties of the trial. Namely the challenges of LARC and LRRC in the pelvis, the importance of R0, the special difficulties in achieving an R0 in the sidewall and posterior compartments, and finally IOERT and the evidence for it. The revised introduction is now just over 1.5 pages as requested.
Comment 2: The authors describe their experience of interdisciplinary discussions on a regional, national and international level: In my opinion, some word should be spent about big data. In modern times, especially when it comes to research in rare cases, physicians / insurance companies and especially national healthcare systems need to have an interest in performing research with big data connecting multiple institutions in order to accumulate patients. What are the intentions of the authors concerning these points? Did the authors opt for multi center approaches for the Electra trial?
Response: We thank the reviewer for this important question. At the inception of the ELECTRA trial there were few standards in the field of advanced high-complexity pelvic exenteration surgery, and no universally accepted surgical terminology or radiological and pathological standards. As a result, no "big data" existed in the field. As mentioned earlier, during the trial's development, we had to address the issue of standards. Consequently, ELECTRA now introduces pathological, surgical, and radiological standards for the first time. These standards are currently being accepted in the UK and are also being collaboratively modified on an international level, with future iterations likely to gain acceptance soon. Once this happens, the concept of accessing big data in this field will become much more feasible. One of the key points we emphasised in the manuscript is that this advancement represents an additional benefit of such trials in moving the field forward.
Reviewer 2 Report
Comments and Suggestions for Authors
In this article, the authors described the designation process of the ELECTRA trial and pointed out obstacles in surgical trials with high complexity and volume operation.
For me it's a novel and interesting topic. The authors clearly described and anticipated difficulties and reasonable solutions to future similar trials.
On the other hand, I am curious if the paper fit the "article" type for this journal. I believe the article would be improved with two aspects. One is to incorporate the detailed design such as outcome assessment, patient number, statistics and rationale. The other aspect is clearer explanation of the feasibility assessment and how/when the trial can be progressed to the next phase.
Author Response
Comment 1: I believe the article would be improved with two aspects. One is to incorporate the detailed design such as outcome assessment, patient number, statistics and rationale.
Response : Thank you for this feedback and we are grateful for the opportunity to improve our manuscript as suggested. Even though we had already included the requested details in section 2.1 of the protocol (lines 227-228) , for ease of access we have now included the relevant details as requested in an additional supplementary file (supplementary file 2).
Comment 2: The other aspect is clearer explanation of the feasibility assessment and how/when the trial can be progressed to the next phase.
Response : Thank you for this comment. We have addressed it in two ways. First, we have added the following sentence in section 2.8.1.: “As a result, it was determined that meeting the aforementioned criteria, coupled with adequate community support, would render a future late-phase study in this field feasible”. Second, in the new Supplementary File 2 you will find the following explanation: "The sample size is based on a 95% confidence interval approach, focused on estimating recruitment to the study. It is estimated that 80 eligible patients will be referred for consideration at the sMDT during the course of the study; this number ensures we will be able to estimate recruitment rate within 11%, sufficient to inform the planning of the future effectiveness study. Assuming just over 50% are eligible and agree to join the study, 42 participants will allow estimation of retention to within approximately 15%, as well as providing information on how IOERT is delivered."
We trust that this response adequately addresses your comment.
Reviewer 3 Report
Comments and Suggestions for Authors
The ELECTRA trial, a three-arm, double-blinded randomized controlled trial, aims to evaluate adding intraoperative electron beam radiotherapy (IOERT) to pelvic exenteration surgery for locally advanced and recurrent rectal cancer. This paper details the significant challenges in designing and implementing such a trial in a low-volume, high-complexity surgical oncology setting, including securing funding, addressing equipoise among participating institutions, ensuring patient and public involvement, and establishing standardized definitions and procedures for surgery, radiology, and pathology. The authors highlight the extensive stakeholder consultations and innovative approaches used to overcome these hurdles, emphasizing the importance of collaboration and multi-level engagement for the successful conduct of complex surgical trials.
The paper presents a valuable contribution to the surgical oncology literature by detailing the practical challenges and innovative solutions employed in the ELECTRA trial. However, minor revisions are suggested to:
- Enhance clarity: Some sections could benefit from improved clarity and flow, particularly regarding the methodology and statistical analysis. Consider restructuring or rephrasing certain passages for better readability.
- Strengthen discussion: The discussion section could be strengthened by more explicitly addressing limitations of the study and comparing the findings to other relevant research more thoroughly. Consider adding a paragraph specifically addressing limitations and future research directions.
These are minor issues that can be readily addressed to enhance the paper's overall impact and quality.
Author Response
Comment 1: Enhance clarity: Some sections could benefit from improved clarity and flow, particularly regarding the methodology and statistical analysis. Consider restructuring or rephrasing certain passages for better readability.
Response: Thank you for your comment. We have revised the manuscript for improved clarity, as requested. These revisions are reflected on pages 3, 5, 9, 12, and 13. Additionally, all detailed methodology and statistical analysis have been confined to Supplementary file 2.
Comment 2: Strengthen discussion: The discussion section could be strengthened by more explicitly addressing limitations of the study and comparing the findings to other relevant research more thoroughly. Consider adding a paragraph specifically addressing limitations and future research directions.
Response: We would like to thank the reviewer for his comments and suggestion and we have now made the recommended change and added a limitations paragraph as requested on pages 12-13 of the revised manuscript.